# Detailed and Average Models of a Grid-Connected MMC-Controlled Permanent Magnet Wind Turbine Generator

**Marwan Rosyadi** [1,*] **, Atsushi Umemura** [2]**, Rion Takahashi** [2] **and Junji Tamura** [2]

[1] Department of Electrical Engineering, Faculty of Engineering, Universitas Muhammadiyah Surabaya, Surabaya 60113, Indonesia

[2] Department of Electrical and Electronic Engineering, Kitami Institute of Technology, Kitami 090-0015, Japan; umemura@mail.kitami-it.ac.jp (A.U.); rtaka@mail.kitami-it.ac.jp (R.T.); tamuraj@mail.kitami-it.ac.jp (J.T.)

\* Correspondence: rosyadi@um-surabaya.ac.id

**Abstract:** In this paper, a detailed model and an average model of an MMC (Modular Multilevel Converter)-controlled Permanent Magnet Synchronous Generator (PMSG)-based direct drive wind turbine are proposed. The models are used to analyze the steady-state and transient characteristics of the grid connectivity study of the wind turbine generator. Configuration of the electrical topology and the control scheme of the wind turbine generator for both models are comprehensively presented. In the detailed model, the MMC circuit is represented by power electronic IGBTs, with switching phenomena considered. Meanwhile, in the average model, the MMC circuit is simplified by using voltage source representation, hence the complexity of the MMC circuit and the simulation duration of the analysis can be reduced. Comparative analysis between the detailed and the simplified models is also investigated through simulation performed using PSCAD/EMTDC. The simulation results show that both models have a good controllability and dynamic stability under steady-state and transient conditions. The simulation results also confirm that the average model has adequate accuracy, and simulation time can be reduced significantly.

**Keywords:** wind turbine generator; modular multilevel converter; permanent magnet synchronous generator; steady-state and transient analyses; grid connectivity study

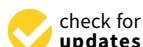



## 1. Introduction

In recent years, wind turbine generators have been significantly increasing in size, in terms of hub height, rotor diameter, and generator capacity, in order to convert more power from wind energy with higher efficiency and lower investment and operating costs. In 2021, the power capacity of wind turbine generators had reached 15 MW [1], and in 2035, the capacity of wind turbine generators is predicted to reach up to 17 MW [2]. Along with the increasing capacity, the connection of the wind turbines to grid power systems by traditional two-level or three-level converters requires multiple electronic devices such as IGBTs in series or parallel connections to achieve very high power capacity and operating voltage. The Modular Multilevel Converter (MMC) is a promising solution for high-power-capacity wind power generation. Compared to other conventional converter technologies, the MMC is a novel converter concept that has many advantages, such as having a simple structure and a flexible design that enables one to expand the number of levels and to replace the submodules easily, so that maintenance becomes easier [3,4]. In addition, connecting the MMC to a grid system without a transformer is possible [5].

The implementation of an MMC in a wind power generator, however, has not been extensively reported. The majority of reports have investigated the stability of grid-connected wind farms using MMC-based High Voltage Direct Current (HVDC) transmission systems [6–10], but only a few papers have discussed the topology concept of the implementation of MMC-controlled wind turbine generators. The performance and topology of MMCs for 2 MW 0.69 kV and 10 MW 10 kV has been discussed in [11]; however, the study only

discusses the grid side converter and concentrates mainly on loss distribution between submodules. In [12], the application of an MMC-controlled multi-phase PMSG was proposed. The fault tolerant control strategy of the MMC is validated via simulation analysis with detailed model representation. However, a comprehensive discussion regarding the modelling of MMC-controlled wind turbine generators for a grid connectivity study has not been widely discussed.

A feasibility study on the stability of wind turbine generators connected to a grid system is particularly important in a wind farm design. The dynamic behavior under steady-state and transient conditions of a wind farm will affect the voltage and frequency stability of the grid system. A large variation in the energy production of a wind farm or lack of synchronism due to a short circuit fault can have a great impact on power system stability and power quality [13,14]. Therefore, modeling and simulation analyses of the grid connectivity of a wind farm in the early stages of development of the facility is essential.

In this paper, detailed and average models of grid-connected MMC-controlled Permanent Magnet Synchronous Generator (PMSG)-based direct drive wind turbines are proposed. The models can be used as representations of the wind turbine generator for the study of grid connectivity. PMSG was chosen on the basis that most manufacturers use this type of generator for their large-capacity wind turbine generators [15–17] due to its high efficiency and attractive features suitable for the wind turbine concept.

In the detailed model, the MMC includes a detailed representation of power electronic IGBT converters, and the MMC circuit is configured by a sub-module that has a two-level half-bridge configuration composed of two IGBTs with anti-parallel diodes and a capacitor. As a switching phenomenon with multi-carrier modulation technique is considered in the modeling, the model should be discretized at a relatively small simulation time step (10 ms). The detailed model is suitable for analyzing the dynamic performance of control systems and harmonics over a short duration.

In the average model, the MMC circuit with power electronic IGBT modules is represented by equivalent voltage sources generating AC voltages. The Switching phenomenon is neglected, and hence this model allows the use of a much larger simulation time step (100 ms). Therefore, the average model is suited for simulation analysis over a longer time.

The organization of the paper can be summarized as follows: Section 2 presents the proposed detailed model of the MMC-controlled PMSG-based wind turbine. Section 3 presents the proposed average model of the MMC-controlled PMSG-based wind turbine. Section 4 discusses the simulation and analysis of a grid-connected wind farm that consists of 5 (five) units of 10 MW PMSG-based wind turbines controlled by an MMC system. The simulation and analysis both focus on the steady-state and transient analyses of both proposed models, and have been performed using PSCAD/EMTDC. Finally, in Section 5, the conclusions of the study are presented.

## 2. Detailed Model of MMC-Controlled Wind Turbine Generator

Figure 1 shows the configuration of the MMC-controlled wind turbine generator in the detailed model. The wind turbine generator is a gearless system in which the wind turbine rotor directly drives the rotor shaft of the PMSG. The PMSG is a multipoles type of generator that operates at variable voltage and frequency. The electrical power produced by the generator is supplied to the grid system with a constant voltage and frequency through a back-to-back converter. The back-to-back converter is formed from two MMCs, namely a stator-side MMC and a grid-side MMC, which are linked by a DC link circuit. The three-phase AC voltage of the PMSG stator winding is rectified by the Stator-side MMC, and the DC voltage of the DC link circuit is inverted to AC voltage by the grid-side MMC. The Stator-side MMC is connected to the stator winding terminal of the PMSG, and the grid-side MMC is connected to the grid system through a step-up transformer (TR). On the stator-side MMC, $I_S^{(abc)}$ and $V_S^{(abc)}$ are, respectively, three-phase currents and voltages from the stator winding, and $\omega_r$ is the rotational speed of the rotor shaft of the PMSG. On the grid-side MMC, $I_G^{(abc)}$ and $V_G^{(abc)}$ are, respectively, three-phase currents and

voltages from the terminal of the grid-side MMC. Each of the MMC systems is equipped with the main MMC controller, inner MMC controller, and the Phase-Shifted Pulse Width Modulation (PS-PWM) circuit. The DC link circuit is configured by two capacitors ($C_{dc}$) arranged in series. The DC link circuit is also equipped with an over-voltage protection system controlled by a DC chopper. For more details, each part of the detailed model of the MMC-controlled wind turbine generator can be explained as follows.

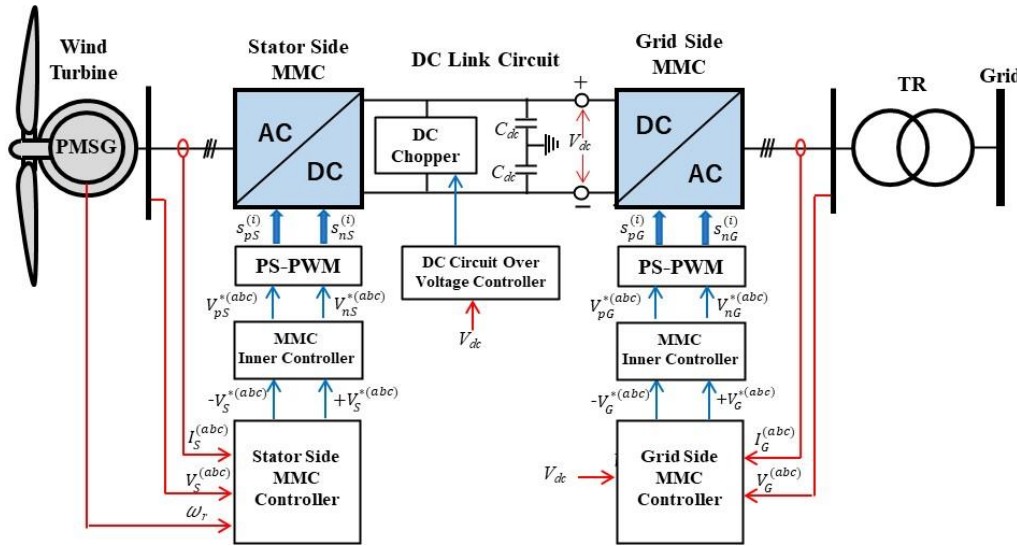

**Figure 1.** Configuration of the MMC-controlled wind turbine generator in the detailed model.

*2.1. Wind Turbine Model*

2.1.1. Power Conversion and Characteristics of Wind Turbine

In this paper, a wind turbine direct driven generator with variable speed concept is considered. The actual mechanical power output of a wind turbine can be written as follows [18]:

$$P_w = 0.5 \, \rho \pi R^2 \, V_w^3 \, C_p(\lambda, \beta) \tag{1}$$

where $P_w$ is the converted power from wind energy (W), $R$ is rotor blade radius (m), $V_w$ is wind velocity (m/s), $\rho$ is air density (Kg/m$^3$), and $C_p$ is power coefficient. The $C_p$ depends on characteristic wind turbine coefficients ($c_1$ to $c_6$), pitch angle ($\beta$), and tip speed ratio ($\lambda$), which can be calculated as follows [16]:

$$C_p(\lambda, \beta) = c_1 \left( \frac{c_2}{\lambda_i} - c_3 \, \beta - c_4 \right) e^{\frac{-c_5}{\lambda_i}} + c_6 \lambda \tag{2}$$

$$\frac{1}{\lambda_i} = \frac{1}{\lambda - 0.08 \, \beta} - \frac{0.035}{\beta^3 + 1} \tag{3}$$

$$\lambda = \frac{\omega_r R}{V_w} \tag{4}$$

The values of the characteristic coefficients of the wind turbine, $c_1$ to $c_6$, are 0.5176, 116, 0.4, 5, 21, and 0.0068, respectively [18], and $\omega_r$ is the rotor speed of the wind turbine (rad/s).

From Equations (1)–(4) the characteristics of the wind turbine are shown in Figures 2 and 3 can be depicted. The relation of power output and rotor speed is depicted in Figure 2, and the relation of power coefficient and tip speed ratio characteristic is depicted in Figure 3. The optimum tip speed ratio ($\lambda_{opt}$) of 8.1 and the optimum power coefficient ($Cp_{opt}$) of 0.48

are obtained when wind velocity is at a rated speed of 12 m/s. The wind turbine power output through Maximum Power Point Tracking (MPPT) is calculated as follows [19]:

$$P_{mppt} = 0.5\rho\,\pi\,R^2\left(\frac{\omega_r R}{\lambda_{opt}}\right)^3 C_{popt} \tag{5}$$

where the wind turbine reference power ($P_{ref}$) is limited to the rated power, 1.0 pu, when the rotor speed is equivalent to or over the rated speed, 1.0 pu.

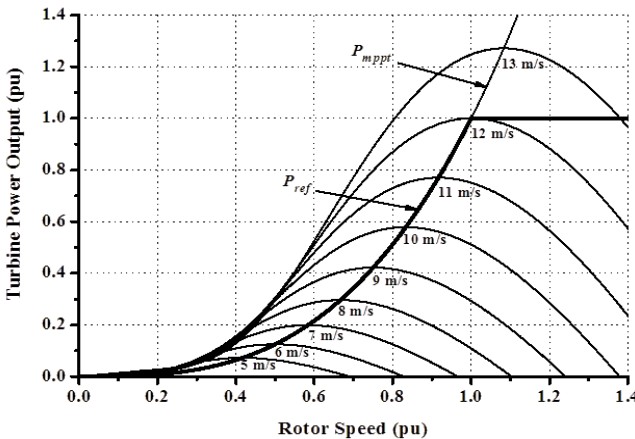

**Figure 2.** Turbine power output vs. rotor speed characteristic.

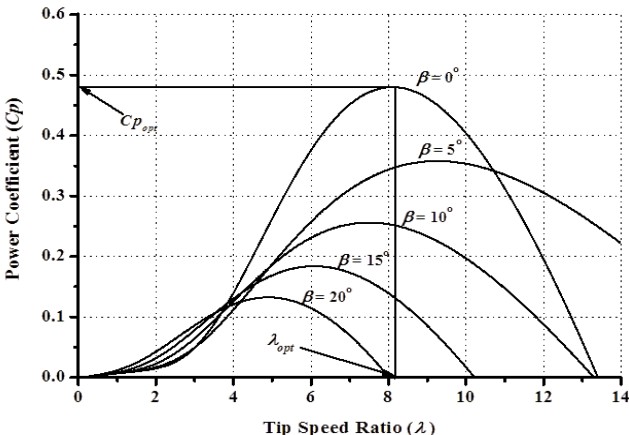

**Figure 3.** Power coefficient vs. tip speed ratio characteristic of wind turbine.

### 2.1.2. Drive Train Model

The moving parts of a direct-driven wind turbine generator are comprised of the following components: rotor blades with pitching mechanism system, a hub, and a rotor shaft. Generally, in the study of the grid connectivity of wind turbine generators, the drive train model is treated as a two-lumped mass or a one-lumped mass model [20]. Because the wind turbine generator in this study is totally seperated from the grid system by a back-to-back MMC system, the one-lumped mass model of the drive train is considered here.

The scheme of the drive train model in the one-lumped mass model is given in Figure 4, which is represented by the following equation [20]:

$$\frac{d\omega_r}{dt} = \frac{T_e - T_m}{J_{eq}} - \frac{B_m}{J_{eq}}\omega_r \tag{6}$$

where $T_m$ is the wind turbine mechanical torque (Nm), $T_e$ is the generator electrical torque (Nm), $J_{eq}$ is the equivalent rotational inertia of the wind turbine generator (kg.m$^2$), and $B_m$ is the damping coefficient (Nm/s), which is derived from:

$$\frac{d\omega_r}{dt} = J_g + \frac{J_t}{n_g^2} \tag{7}$$

where $J_g$ and $J_{wt}$ are, respectively, the rotational inerias of the generator rotor and the wind turbine rotor, and $n_g$ is the gear ratio, which can be set at 1 in the case of a direct drive system (without gearbox).

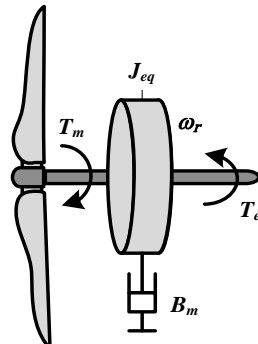

**Figure 4.** One-lumped mass model of wind turbine generator.

### 2.1.3. Pitch Blade Controller

The power output of the wind turbine generator always fluctuates depending on wind speed variations, and its power output is not allowed to exceed its rating capacity. Therefore, the pitch blade controller works to maintain the rotor speed of the wind turbine so as not to exceed the permissible speed limit. The schematic diagram of the pitch blade controller model is shown in Figure 5 [21]. The controller maintains the rotor speed of the wind turbine ($\omega_r$), ensuring that it does not exceed its reference value ($\omega_r^*$). The control loop of the pitch actuator is represented by a first-order transfer function with time constant $T$. The PI controller is used to obtain a pitch-angle reference ($\beta^*$).

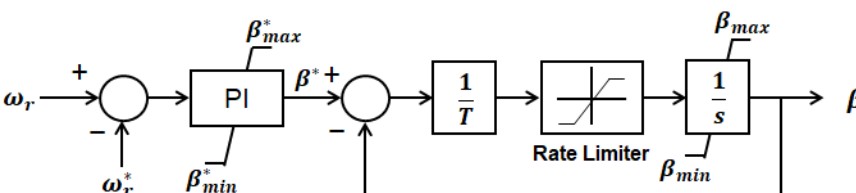

**Figure 5.** The pitch blade controller model.

### 2.2. Generator Model

For the generator model, the permanent magnet synchronous machine model in the PSCAD/EMTDC master library is considered in this study. Voltage equations for the main stator windings, voltage equations for the short-circuited windings, and the flux linkage equations of the windings are represented in the *dq0* reference frame. A more detailed explanation of the generator in the PSCAD/EMTDC model can be found in [22,23].

### 2.3. MMC System
#### 2.3.1. Configuration and Operation of MMC

The configuration of the three-phase MMC system is depicted in Figure 6. The MMC in this study is a 7 (seven)-level modular converter. The MMC consists of three-phase legs (Leg A, Leg B, and Leg C). The legs have two similar arms, i.e., upper arm (p) and

lower arm (n). Each arm consists of six identical sub-modules (SM), an arm inductor ($L_{arm}$), and an arm resistance ($R_{arm}$), arranged in series. The arm inductors are used to limit circulating arm current between the three-phase units and the valve short circuit current, and to contribute to interface between the AC grid system and the MMC. The sub-module has a two-level half-bridge configuration composed of two IGBTs with anti-parallel diodes and a capacitor ($C_{SM}$). Three-phase AC terminal voltage is connected to each phase leg on the common point connection between the upper and lower arms through a phase reactor consisting of an inductor ($L_f$) and a resistor ($R_f$). The reactor is used as a filter for AC voltage and current [24].

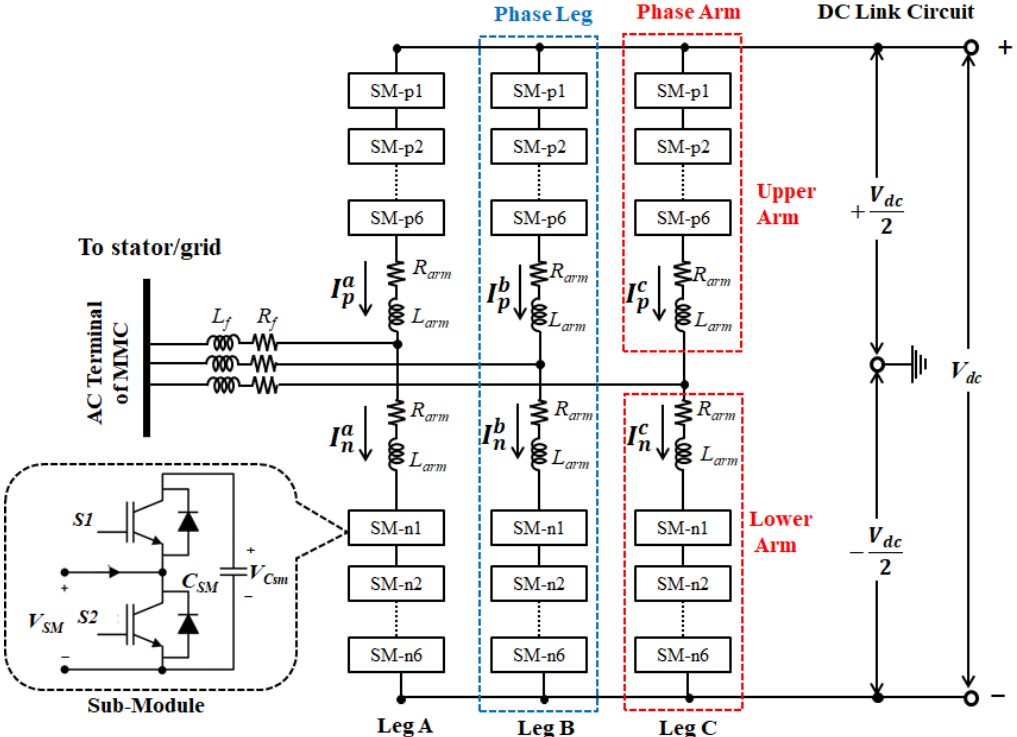

**Figure 6.** Configuration of MMC.

In the operation of the MMC, the DC link voltage ($V_{dc}$) charges the capacitor ($C_{SM}$) in the entire sub-module, in which the sub-modules are switched into an inserted state or a bypassed state. In the inserted state, the sub-module capacitor can be charging or discharging, depending on the voltage reference polarity. The switching state of the sub-module conditions is given in Table 1. Figure 7 shows the direction of sub-module arm current ($I_{arm}$) according to S1 and S2 switching state. The positive polarity of the arm current direction is indicated in red, and the negative polarity of the arm current direction is indicated in blue.

**Table 1.** Switching state of the Sub-Module conditions.

| Switching State | | Terminal Voltage of SM ($V_{SM}$) | Arm Current Polarity | Status of Capacitor |
|---|---|---|---|---|
| **S1** | **S2** | | | |
| OFF | ON | 0 | + | Bypass |
| ON | OFF | $Vc_{sm}$ | + | Charging |
| OFF | ON | 0 | − | Bypass |
| ON | OFF | $Vc_{sm}$ | − | Discharging |

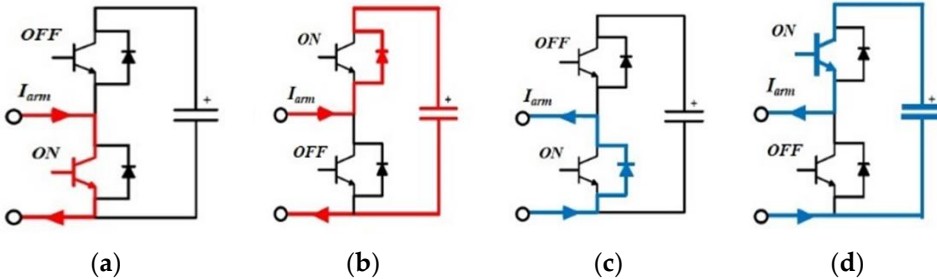

**Figure 7.** Current flow in the sub-module according to switching state: (**a**) Bypass in positive polarity (**b**) Charging in positive polarity (**c**) Bypass in negative polarity (**d**) Discharging in negative polarity.

### 2.3.2. Stator-Side MMC Controller

The active power and the reactive power output of the PMSG is controlled by the stator-side MMC controller. The tree-phase current and the voltage of the stator winding are transformed into the *dq*-axis components using Park Transformation, where the position of the rotor angle ($\theta_r$) is obtained from the rotational speed of the PMSG. Details of the control scheme of the stator-side MMC controller is shown in Figure 8. The active power and reactive power are controlled independently by the *q*-axis current component and the *d*-axis current component, respectively. The PI controllers adjust the power loop and inner current loop controllers for each of the *d*-axis and the *q*-axis components. The generator (PMSG) usually operates in unity power factor operation, in which the active power output ($P_S$) is set according to power reference ($P_{ref}$) tracking by the MPPT circuit, and the reactive power output ($Q_S$) is set at zero. The outer power loop controller generates the dq-axis reference currents ($I_S^{*(dq)}$). The inner current loop controller generates the dq-axis reference voltages (($V_S^{*(dq)}$). By using invers Park Transformation, the *dq*-axis reference voltages are transferred into sinusoidal three-phase reference voltages ($+/-V_S^{*(abc)}$) in which the minus ($-$) and plus ($+$) indicate the reference signal for the upper arm and the lower arm, respectively. The Park Transformation equations can be found in Appendix A. To increase the tracking capability of the controllers, the cross-coupling cancellation $\omega_r\left(L_{Sarm}/2 + L_{Sf} + L_S^{(dq)}\right)$ is added at the output of the inner loop of the current controller. $L_S^{(dq)}$ denotes the *dq*-axis components of leakage inductance of the stator winding of the generator, and $L_{Sarm}$ and $L_{Sf}$ denote the arm inductance and reactor inductance of the stator-side MMC, respectively.

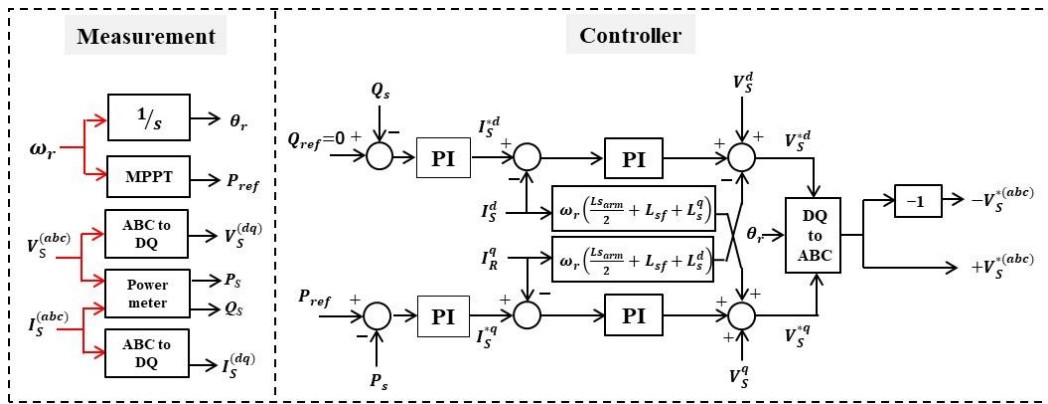

**Figure 8.** Stator-Side MMC Controller.

### 2.3.3. Grid-Side MMC Controller

The control scheme of the grid-side MMC controller is shown in Figure 9. The control loop is designed for controlling the DC link circuit voltage ($V_{dc}$) and the reactive power output of the grid-side MMC ($Q_g$) in a similar way to the stator-side MMC based on the *dq*

vector control. Three-phase currents ($I_G{}^{(abc)}$) and voltages ($V_G{}^{(abc)}$) at the AC terminal of the grid-side MMC are transformed into the *dq*-axis form using Park Transformation, where the phase angle ($\theta_G$) and angular frequency ($\omega_G$) are obtained from the Phase Locked Loop (PLL) controller. The PLL controller technique used in this study refers to the original PSCAD/EMTDC's master library. The instantaneous active power ($P_G$) and reactive power ($Q_G$) are calculated using the power meter, and the instantaneous rms voltage ($V_G$) of the AC terminal is obtained from the three-phase rms voltmeter.

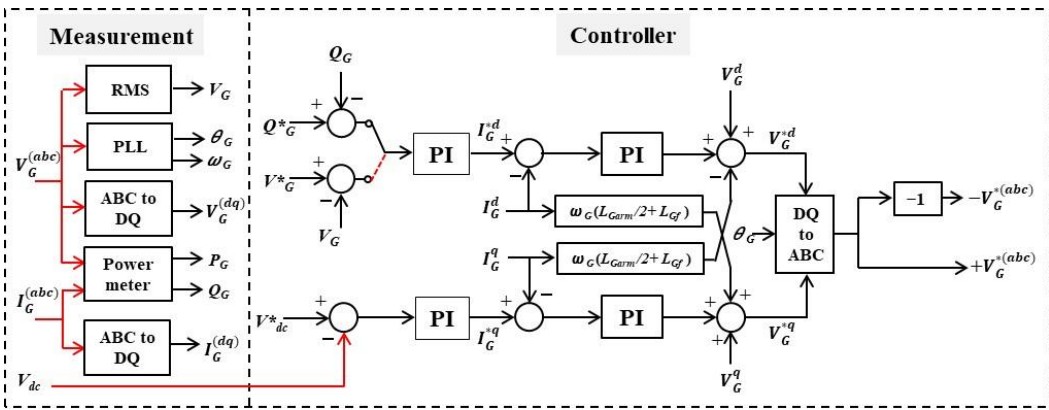

**Figure 9.** Grid-side MMC Controller.

In the grid-side MMC controller, the q-axis component $\left(I_G^q\right)$ is used to control DC link circuit voltage at a constant DC voltage reference ($V_{dc}^*$), and the *d*-axis current component $\left(I_G^d\right)$ is used to control the reactive power output ($Q_G$) or the AC terminal voltage ($V_G$) of the grid-side MMC. In normal mode operation, the reactive power reference ($Q_G^*$) is usually set at zero to maintain the power factor at unity. In fault mode operation, the grid voltage reference ($V_G^*$) is set at 1.0 pu. During fault conditions, the grid-side MMC controller will change its operation from normal mode to fault mode when the voltage at the AC terminal output drops below 80%. The PI control loop systems also consist of the inner current loop control and the power loop control for each of the *d*-axis and the *q*-axis components. The cross coupling in the term of $\omega_G\left(L_{Garm}/2 + L_{Gf}\right)$ is added to the output of the inner loop current control for the controller tracking improvement, where $L_{Garm}$ and $L_{Gf}$ are the arm inductance and the reactor inductance of the grid-side MMC, respectively.

The aim of the development of the wind turbine generator models in this paper is to simulate and analyze the dynamic behaviors of a grid-connected wind farm under steady-state and transient conditions. The grid connectivity study is an important requirement in the development stage of any wind farm project. In order to improve the dynamic performance of the control system of the stator-side MMC and grid-side MMC, improvement methods such as energy-shaping L2-gain [25], sliding mode controller [26], pole and placement, etc., can be applied to the proposed model. However, because our only concern in this study is grid connectivity, the control system that is applied to the stator-side MMC and the grid-side MMC can be a standard control model using a PI controller. The gain control parameters, such as *Kp* and *Ki*, are obtained using a pole and placement method, and an optimum symmetrical criterion method for inner and outer controllers, respectively.

### 2.3.4. MMC Inner Control

As the MMC has three-phase legs, a circulating current can exist within each phase leg. In addition, a two-arm configurations in the MMC can generate two different voltage levels for each sub-module in the same arm [27]. The circulation phase leg current should be eliminated, and each sub-module capacitor voltage should be kept balanced at the same level. To handle these problems, the MMC inner controller is introduced. The MMC inner controller depicted in Figure 10 is applied to both the stator-side MMC and the grid-side

MMC. The purpose of the MMC inner controller is to control circulation current in the phase legs at zero, and to maintain each sub-module capacitor voltage at the same level. The average circulation current through a phase leg ($I_{diff}$) is obtained by adding the current through the upper arm ($I_p$) to the current through the lower arm ($I_n$) divided by 2. The circulating current reference for each phase leg is set at zero. The PI controller adjusts to impose the circulation voltage ($V_{diff}$) into the reference voltage for the upper arm and the lower arm. To control the voltage balancing of each SM capacitor, extra controllers are required. A detailed explanation for the circulating current controller and the SM capacitor voltage balancing controller can be found in [28].

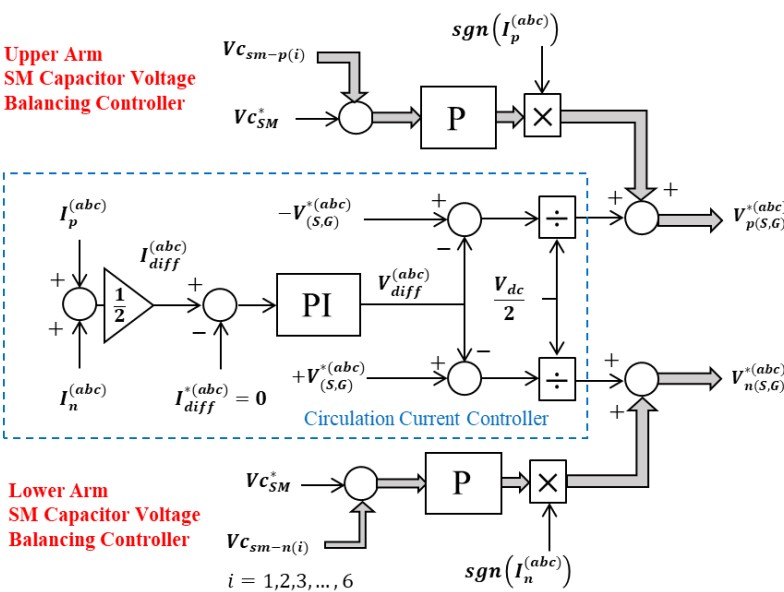

**Figure 10.** MMC Inner Controller.

### 2.3.5. Phase Shifted Pulse Width Modulation (PS-PWM)

In general, modulation techniques for multilevel converters can be summarized in three categories: Multi Carrier Pulse Wave Modulation (MC-PWM), Nearest Level Modulation (NLM), and Space Vector Modulation (SVM) [29,30]. The multi-carrier Phase Shifted Pulse Wave Modulation (PS-PWM) technique is considered in this study due to its merits compared to other approaches. The PS-PWM technique is more effective and superior in controlling the MMC, that is, the power distribution over the entire sub-modules can be provided, and voltage balance at the sub-module capacitors can be achieved. Figure 11 shows the process of the PS-PWM technique. Each sub-module has an independent carrier signal in which the reference signal is distributed to all the series sub-modules in each leg. The goal of the modulation is to produce a PWM signal for switching the IGBT gate on each sub-module. The number of the carrier signal applied depends on the level of the MMC, and in this case, it is N−1, where N is the level of the MMC. The phase shift ($\phi$) between the carrier signals can be obtained through $\phi = 360^{\circ}/(N - 1)$ [31]. The frequency and amplitude of all carrier signals should be equal. As the SM capacitor voltage balancing controller is applied to each SM, the PS-PWM with individual capacitor voltage control technique [28] is considered.

### 2.4. DC Link Circuit and Overvoltage Protection System

Figure 12 shows the DC link circuit model of a back-to-back MMC converter. The circuit model consists of two DC link capacitors, a DC chopper, and an overvoltage protection controller. The DC link circuit is a connection circuit that connects the stator-side MMC and the grid-side MMC. The DC voltage on the DC link circuit should be kept constant at the rated operating DC voltage so that the power flow from the generator to the grid system can be achieved smoothly. When a disturbance such as a short circuit occurs in the

grid system, the active power on the AC terminal of the grid-side MMC decreases, while the active power from the generator is still produced. This condition leads to a significant increase of the DC link circuit voltage due to a power imbalance between the stator-side MMC and the grid-side MMC. When the DC voltage is larger than 1.05 pu, the overvoltage protection controller is activated, and then the active power from the generator is absorbed by a chopper resistance ($R_{ch}$). The value of the resistance can be adjusted according to the amount of active power produced by the wind turbine generator, which is represented by the reference power ($P_{ref}$).

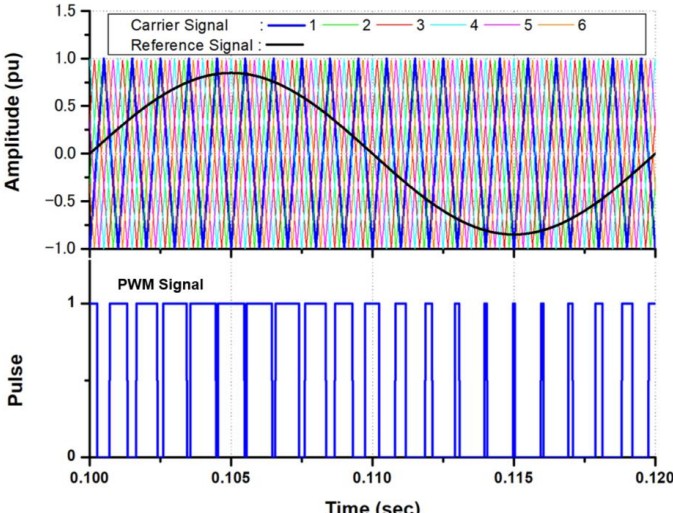

**Figure 11.** PS-PWM Technique.

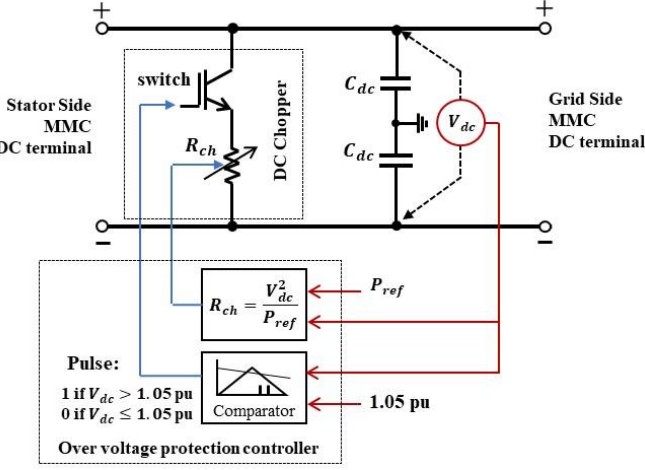

**Figure 12.** DC link Circuit model.

## 3. Average Model of MMC-Controlled Wind Turbine Generator

The aim of the average model is to reduce the complexity and simulation time. The study of grid-connected wind farms with many generators becomes inefficient if the simulation is performed using the detailed model. In the average model, the power electronic IGBTs of the MMC circuit and switching phenomenon are neglected, and hence the model become simpler. The configuration of the average model of the MMC-controlled wind turbine generator is shown in Figure 13. The main part of the model consists of a wind turbine generator, including a pitch controller system, stator-side MMC with a controller system, a DC link circuit including overvoltage protection, and a grid-side MMC with a controller system. The wind turbine, drive train, pitch controller, and generator models are the same as those used in the detailed model. Likewise, the stator-side MMC

controller and the grid-side MMC controller are the same as the controllers used in the detailed model. Only the MMC circuits and the DC link circuit are simplified. It should be noted that the behavior of the sub-module capacitor voltage is not considered in the average model. Therefore, the inner MMC controller can be omitted.

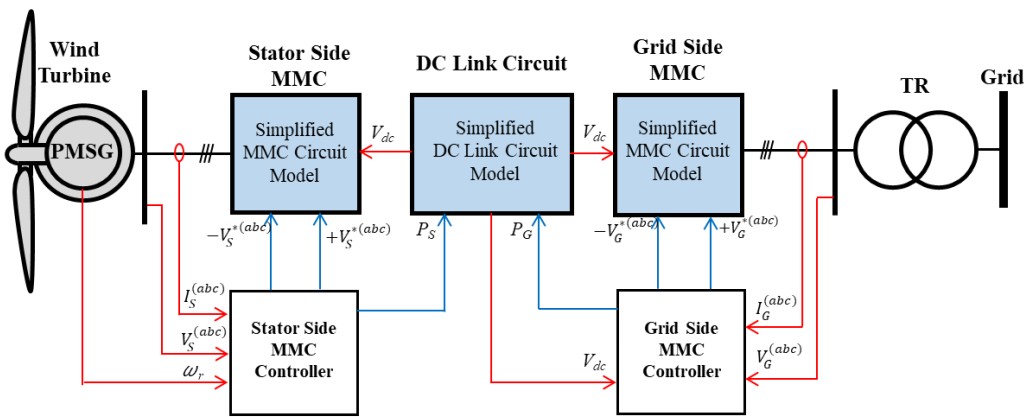

**Figure 13.** Average model configuration of a wind turbine generator.

### 3.1. MMC Circuit Model

To derive a model representation of the MMC, the equivalent approach model, as shown in Figure 14, is considered. By applying Kirchhoff Voltage Low (KVL) across the phase reactor, differential equations for the three-phase circuit can be expressed as follows [32,33]:

$$L_f \frac{di_f^{(abc)}}{dt} = v_g^{(abc)} - v_c^{(abc)} - R_f i_f^{(abc)} \tag{8}$$

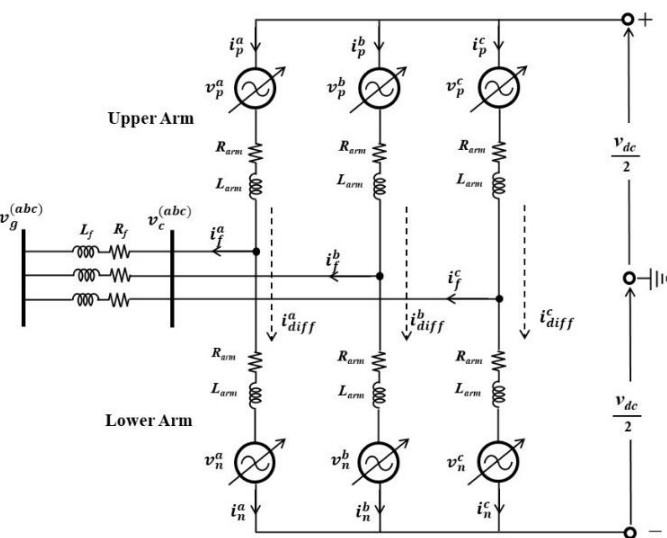

**Figure 14.** Electrical circuit model of MMC.

By applying Park Transformation, the three-phase differential equation in the *dq*-axis component can be expressed as follows:

$$L_f \frac{di_f^d}{dt} = v_g^d - v_c^d - R_f i_f^d + \omega_g L_f i_f^q \tag{9}$$

$$L_f \frac{di_f^q}{dt} = v_g^q - v_c^q - R_f i_f^q - \omega_g L_f i_f^d \tag{10}$$

The complex grid power ($S_g$) can be calculated as follows:

$$S_g = \left(v_g^d + jv_g^q\right)\left(i_f^d - ji_f^q\right) \Rightarrow S_g = \left(v_g^d i_f^d + v_g^q i_f^q\right) + j\left(v_g^q i_f^d - v_g^d i_f^q\right) \tag{11}$$

From (11), the active power and reactive power can be written as:

$$P_g = v_g^d i_f^d + v_g^q i_f^q \tag{12}$$

$$Q_g = v_g^q i_f^d - v_g^d i_f^q \tag{13}$$

In the same way, active power and reactive power flow to the converter valve can be written as:

$$P_c = v_c^d i_f^d + v_c^q i_f^q \tag{14}$$

$$Q_g = v_c^q i_f^d - v_c^d i_f^q \tag{15}$$

As reactive power does not propagate to the DC side of the converter valve, the DC current ($i_{dc}$) is obtained with regards to the active power balance between the AC side and the DC side. By assuming that the losses on the converter can be omitted, the following relation can be written:

$$P_c = P_{dc} \Rightarrow v_c^d i_f^d + v_c^q i_f^q = v_{dc} i_{dc} \tag{16}$$

The voltages waveform at the AC side of the MMC depends on the reference voltages fed to each of the six arms. The inserted voltages in the six arms of the MMC are represented by a controlled voltage source. The upper and lower arm currents can be written as follows:

$$i_p^{(abc)} = i_{diff}^{(abc)} + \frac{i_f^{(abc)}}{2} \tag{17}$$

$$i_n^{(abc)} = i_{diff}^{(abc)} - \frac{i_f^{(abc)}}{2} \tag{18}$$

where $i_{diff}$ is the circulating phase leg current, which can be determined as the average of the upper arm and lower arm currents as:

$$i_{diff}^{(abc)} = \frac{i_p^{(abc)} + i_n^{(abc)}}{2} \tag{19}$$

The voltage output at the AC side of the MMC is given by:

$$v_c^{(abc)} = \frac{v_p^{(abc)} - v_n^{(abc)}}{2} - \frac{R_{am}}{2}i_f^{(abc)} - \frac{L_{am}}{2}\frac{di_f^{(abc)}}{dt} \tag{20}$$

The DC loop of each MMC arm can be expressed as:

$$L_{arm}\frac{di_{diff}^{(abc)}}{dt} + R_{arm}i_{diff}^{(abc)} = \frac{v_{dc}}{2} - \frac{v_p^{(abc)} + v_n^{(abc)}}{2} \tag{21}$$

The inner difference voltage of each phase is given by:

$$v_{diff}^{(abc)} = L_{arm}\frac{di_{diff}^{(abc)}}{dt} + R_{arm}i_{diff}^{(abc)} \tag{22}$$

The reference voltage of the upper arm and the lower arm for each phase can be written as follows:

$$v_{pref}^{(abc)} = \frac{v_{dc}}{2} - \frac{v_p^{(abc)} - v_n^{(abc)}}{2} - v_{diff}^{(abc)} \tag{23}$$

$$v_{nref}^{(abc)} = \frac{v_{dc}}{2} + \frac{v_p^{(abc)} - v_n^{(abc)}}{2} - v_{diff}^{(abc)} \tag{24}$$

Referring to Figure 14, the proposed simplified MMC circuit model can be represented as depicted by Figure 15. The equivalent circuit for the MMC is represented by a pair of three-phase AC voltage sources and a pair of DC voltage sources for the upper arm and the lower arm, respectively. The AC voltage sources are connected to the MMC AC terminal through the arm inductor ($L_{arm}$), arm resistance ($R_{arm}$), reactor inductor ($L_f$), and reactor resistance ($R_f$). The series DC voltage sources are connected to the DC terminal of the MMC; $-V_{S,G}^{(abc)}$ and $+V_{S,G}^{(abc)}$ are the reference voltages for the upper and lower arms, respectively, from the MMC controller. $V_{dc}/2$ is used as the reference voltage for the DC voltage sources, where $V_{dc}$ is the voltage from the DC link circuit.

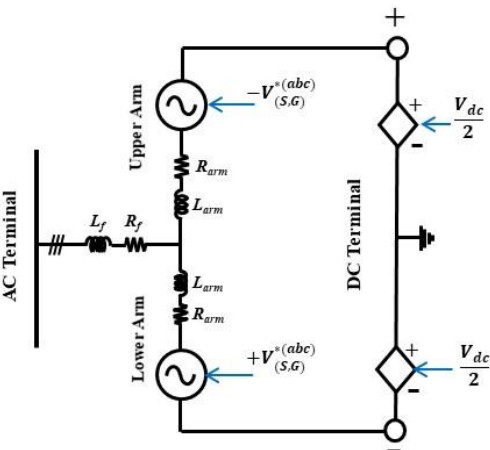

**Figure 15.** Simplified MMC circuit model.

### 3.2. DC Link Circuit Model

The dynamic behavior of the capacitor voltage can be expressed by the following equation [34]:

$$\frac{dV_{dc}}{dt} = \frac{1}{C_{dc}V_{dc}}(P_S - P_G - P_{ch}) \tag{25}$$

where $P_{ch}$ is the power absorbed by chopper resistance.

In steady-state condition, the DC link circuit voltage should be kept constant at the rated voltage, and hence the power produced by the generator can flow to the grid. When a transient disturbance, such as a short circuit, occurs in the grid system, the DC link circuit voltage can exceed its rated voltage significantly due to the imbalance of output power between the PMSG ($P_S$) and the grid-side MMC ($P_G$). This phenomenon is important to consider in any grid connectivity study of wind farms. Therefore, the overvoltage protection scheme is also included in the model. The configuration of the proposed DC link circuit model is shown in Figure 16. Power absorption by the chopper resistance is activated when the DC link voltage exceeds 1.05 pu.

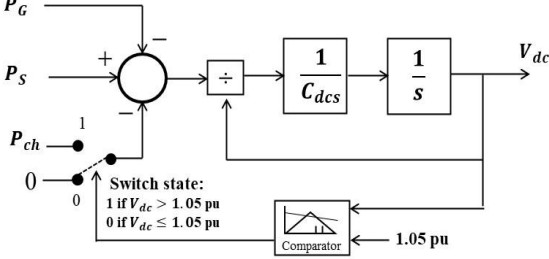

**Figure 16.** DC link circuit model.

## 4. Simulation and Analysis

Figure 17 depicts the power system considered in the simulation study: a 50 MW wind farm consisting of five 10 MW PMSG wind turbines controlled by an MMC system. The wind turbine generators (WTGs) are connected to each other by medium voltage collector power cables. Each collector power cable is represented by an equivalent π circuit model. The power outputs from the wind turbine generators are collected at a 33 kV main bus (B33) and then supplied to the main grid through a 33 kV/66 kV main transformer and double circuit 66 kV transmission line. The parameters of the PMSG and the MMC are presented in Tables 2 and 3, respectively.

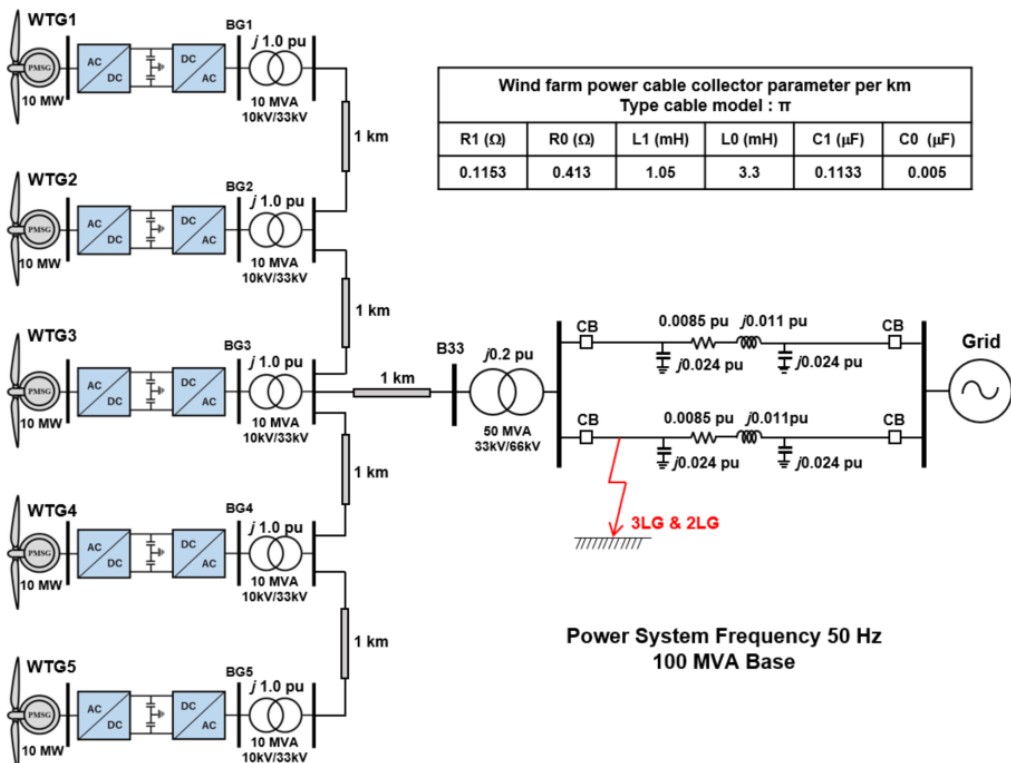

**Figure 17.** Power system model.

The power system model shown in Figure 17 has been analyzed with the detailed and the average models using PSCAD/EMTDC. The accuracy of the average model has been validated by comparing its responses under both steady-state and transient conditions with those obtained from the detailed model. The simulations have been performed on a personal computer (Intel (R) Core (TM) i7-9700 CPU @ 3.0 GHz Ram 64 GB).

### 4.1. Steady-State Performance Analysis

In the steady-state study, natural wind velocity data measured at Hokkaido Island, Japan, are randomly selected for the simulation. A wind speed of 300 s was applied to each wind turbine generator, and the results are shown in Figure 18. In this paper, the dynamic performances of all of the wind turbine generators are not shown; only the dynamic performances of WTG1 are presented as a representative for all WTGs. The dynamic performances of WTG 1, such as rotor speed response, the voltage profile at the terminal of the stator winding, active and reactive power output of the PMSG, the DC voltage profile at the DC link circuit, active and reactive power output of the grid-side MMC, and voltage profile at the low voltage side of the step-up transformer (Bus BG1), are presented in Figure 19. The total active power and the reactive power output of the wind farm and voltage profile at Bus B33 are presented in Figure 20. In can be confirmed that the dynamic performances of the WTG under steady-state condition can be clearly analyzed

in the detailed model and the average model representation, and good dynamic control performances can clearly be seen. It is clear that the average model has sufficient accuracy for steady-state analysis. Moreover, the computation time of the average model is much shorter than that of the detailed model, as presented in Table 4.

**Table 2.** Parameters of PMSG.

| Parameter | Value | Parameter | Value |
| --- | --- | --- | --- |
| Rated MVA | 10 MVA | Stator Winding Resistance | 0.017 pu |
| Rated Voltage (L-L) | 10 kV | Stator Leakage Reactance | 0.0364 |
| Rated Frequency | 50 Hz | *d-axis* Unsaturated Reactance $[X_d]$ | 0.55 pu |
| Magnetic Strength | 1.1 pu | *q-axis* Unsaturated Reactance $[X_q]$ | 1.11 pu |

**Table 3.** Parameters of MMC.

| Parameter | Value | Parameter | Value |
| --- | --- | --- | --- |
| Rated Power | 10 MVA | SM Capacitor | 9.2 mF |
| Rated AC Voltage | 10 kV | Arm Inductance | 3.8 mH |
| Rated DC Voltage | 18 kV | Arm Resistance | 0.097 Ω |
| Number of SMs per arm | 6 | Reactor Inductance | 0.773 mH |
| Carrier Frequency | 1000 Hz | Reactor Resistance | 0.019 Ω |
| DC link Capacitor | 7.4 mF | | |

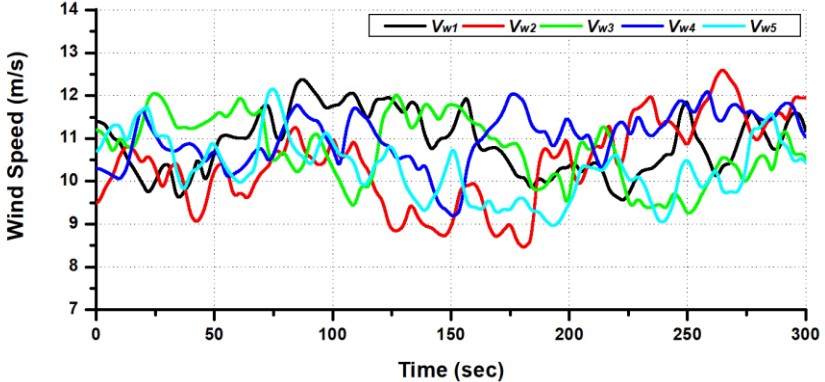

**Figure 18.** Wind velocity data.

*4.2. Transient Performance Analysis*

For the transient performance study, a short circuit of three lines to ground fault (3LG) and two lines to ground fault (2LG) on one of the circuits of the 66-kV transmission line is considered as disturbance. The location of the faults is shown in Figure 17. The faults occur at 0.1 s, and then the circuit breakers (CBs) on the faulted line are opened at 0.2 s to isolate the fault from the system. The CBs are reclosed at 1.0 s on the assumption that the fault has been cleared. During the simulation time of 5 s, the wind velocities of the WTGs are assumed to be constant at a rated velocity of 12 m/s.

As the wind speed applied to each wind turbine generator is the same, the dynamic performances of each individual WTG in the transient conditions for 3LG and 2LG are represented by the dynamic performances of only WTG1, as presented in Figures 21 and 22, respectively. As the wind turbine generator is totally decoupled from the grid network by the back-to-back MMC, the short circuit on the grid network does not affect dynamic performances on the generator side in cases of 3LG and 2LG. The transient disturbances have almost no influence on the dynamic performances of the generator, such as rotor speed response, power outputs, and voltage profile. However, the transient disturbances affect the performances of the DC link circuit voltage, the power output of the grid-side

MMC, and the current and voltage profiles at Bus BG1. In the DC link circuit, a transient DC voltage appears due to the 3LG and 2LG faults, but it can be controlled by the overvoltage protection system, and hence the DC voltage can be returned to the initial condition after the fault is cleared. It is also confirmed that the power output of the grid-side MMC and the voltage profile at Bus BG1 can be returned to the initial conditions after the fault. The performances of the total power output of the wind farm and the voltage profile at Bus 33 for the 3LG and 2LG cases are shown in Figures 23 and 24, respectively.

From the simulation results, it can be confirmed that the dynamic performances of the detailed model and the average model have almost the same responses under the transient conditions. The different responses between the detailed and the average models appear in the DC link circuit voltage profile and the power output of the grid-side MMC. However, they are slight and disappear after returning to the steady-state condition after fault clearance.

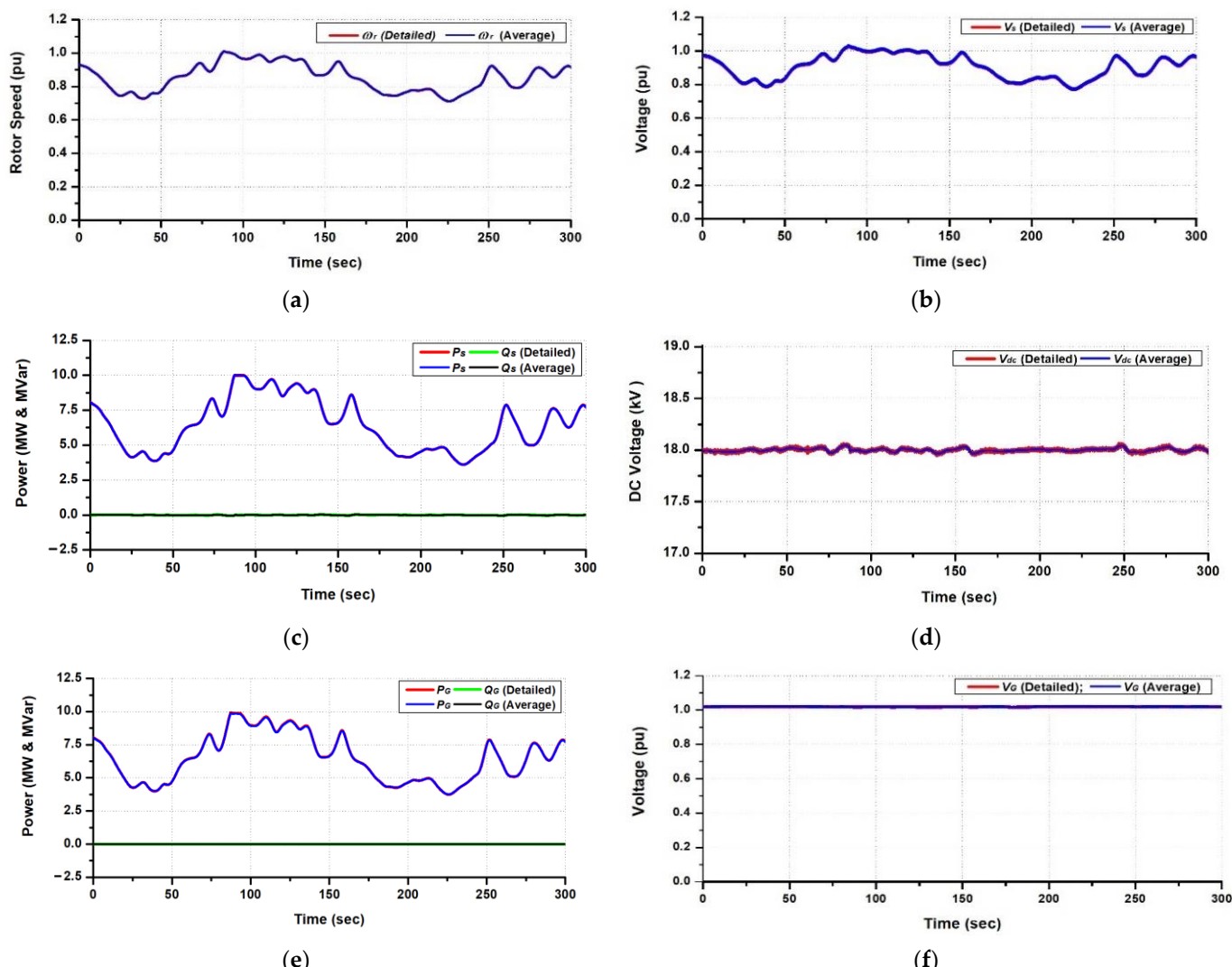

**Figure 19.** Dynamic Performances of WTG1 under steady-state condition: (**a**) Rotor speed response; (**b**) Voltage profiles at the terminal of the stator winding; (**c**) Active and reactive power output of the PMSG; (**d**) DC link circuit voltage profile; (**e**) Active and reactive power output of grid-side MMC; (**f**) Voltage profile at Bus BG1.

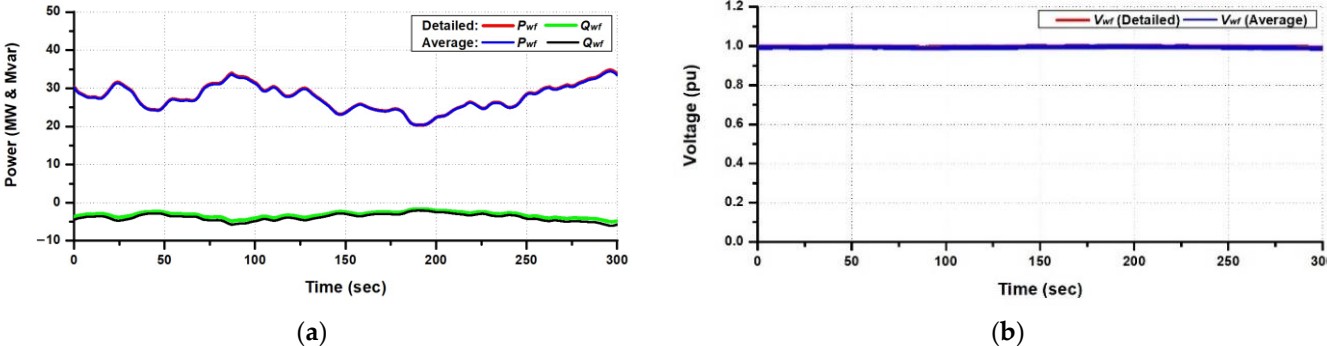

(**a**) (**b**)

**Figure 20.** Power output and voltage performances of the wind farm under steady-state condition: (**a**) Total active power and reactive power outputs of the wind farm; (**b**) Voltage profile at Bus B33.

**Table 4.** Computation time of each model for 300 sec steady-state analysis.

| Simulation | Computation Time | |
|---|---|---|
| | **Detailed Model** | **Average Model** |
| Time step | 10 μs | 100 μs |
| Duration | 840 h | 11 min |

### 4.3. Discussion

The purpose for the development of the wind turbine generator models in this paper was to simulate and analyze the dynamic behaviors of a grid-connected wind farm under steady-state and transient conditions. To verify the accuracy of the proposed models, the simulation results should be validated by real field data. However, real field data is not easy to access. In addition, the implementation of an MMC-controlled PMSG-based wind turbine is a new topology concept for a wind turbine generator that has not yet been realized practically. Therefore, the detailed model is presented in this study because its dynamic behavior is close to real field conditions. To obtain a real system situation, the wind farm model shown in Figure 17 is adopted and simulated as the proposed detailed model. As a consequence, the simulation involved extensive computation time.

It is very important to understand that the proposed average model is an approach method to approximate the detailed model by reducing the complexity of the MMC circuit, where some parts, such as the switching phenomena and power electronic IGBTs of the submodules, are omitted in the simulation. In the steady-state analysis, both models have almost the same responses. In the case of a transient condition with a short circuit fault, there are slight differences in the simulation results in the DC voltage response and the active and reactive power response overshoot between the detailed model and the average model after the CB is opened.

As a real wind farm can consist of hundreds of wind turbine generators, the simulation study cannot be performed by the detailed model. To solve this problem, the average model is presented in this paper. The average model can be used as an individual or an aggregated wind turbine generator models in simulation analyses.

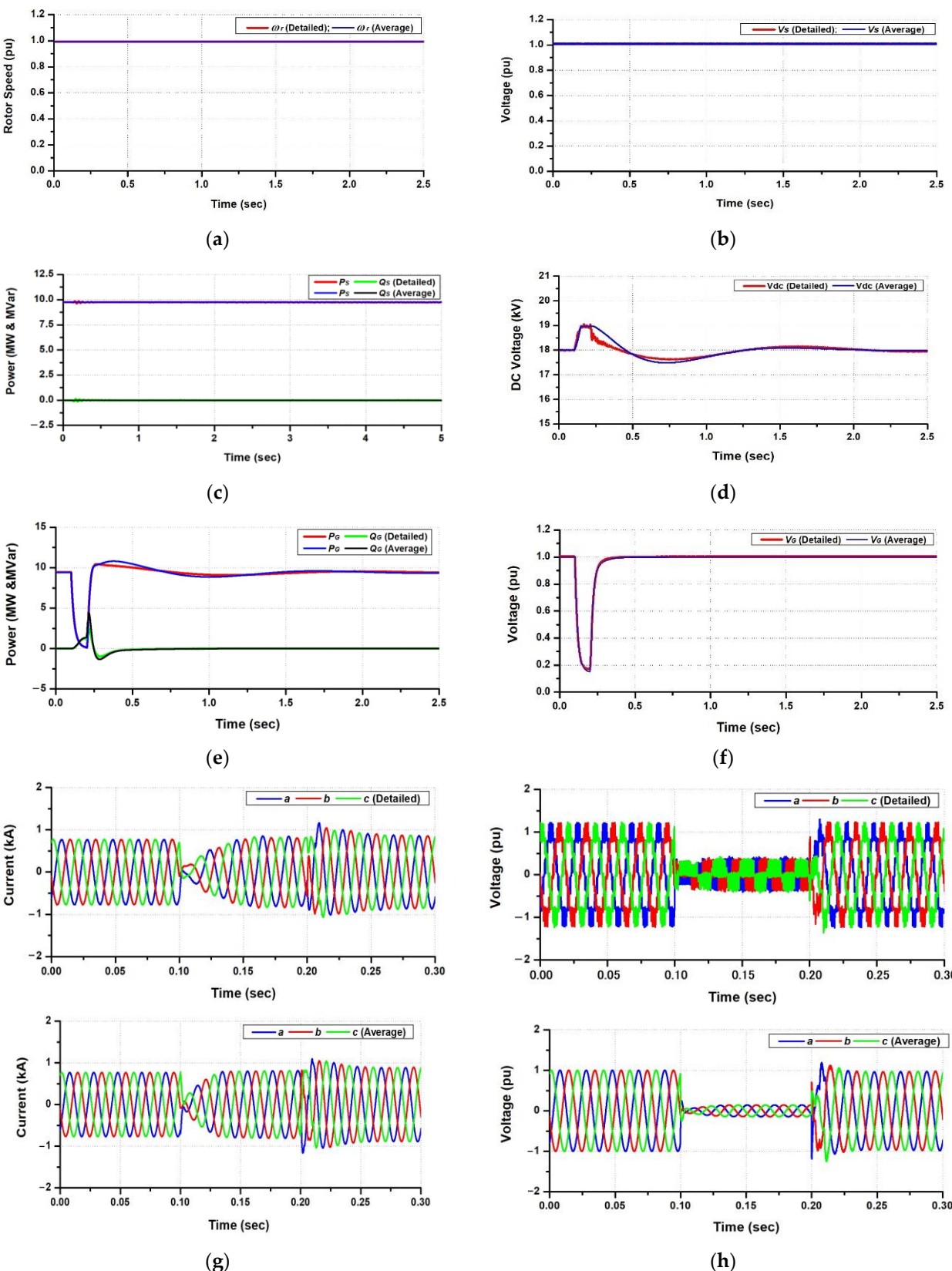

**Figure 21.** Dynamic performances of WTG1 in the case of 3LG: (**a**) Rotor speed response; (**b**) Voltage profiles at the terminal of the stator winding; (**c**) Active and reactive powers output of the PMSG; (**d**) DC link circuit voltage profile; (**e**) Active and reactive power output of grid-side MMC; (**f**) Voltage profile at Bus BG1; (**g**) Waveform of current at the grid-side MMC; (**h**) Waveform of voltage at the grid-side MMC.

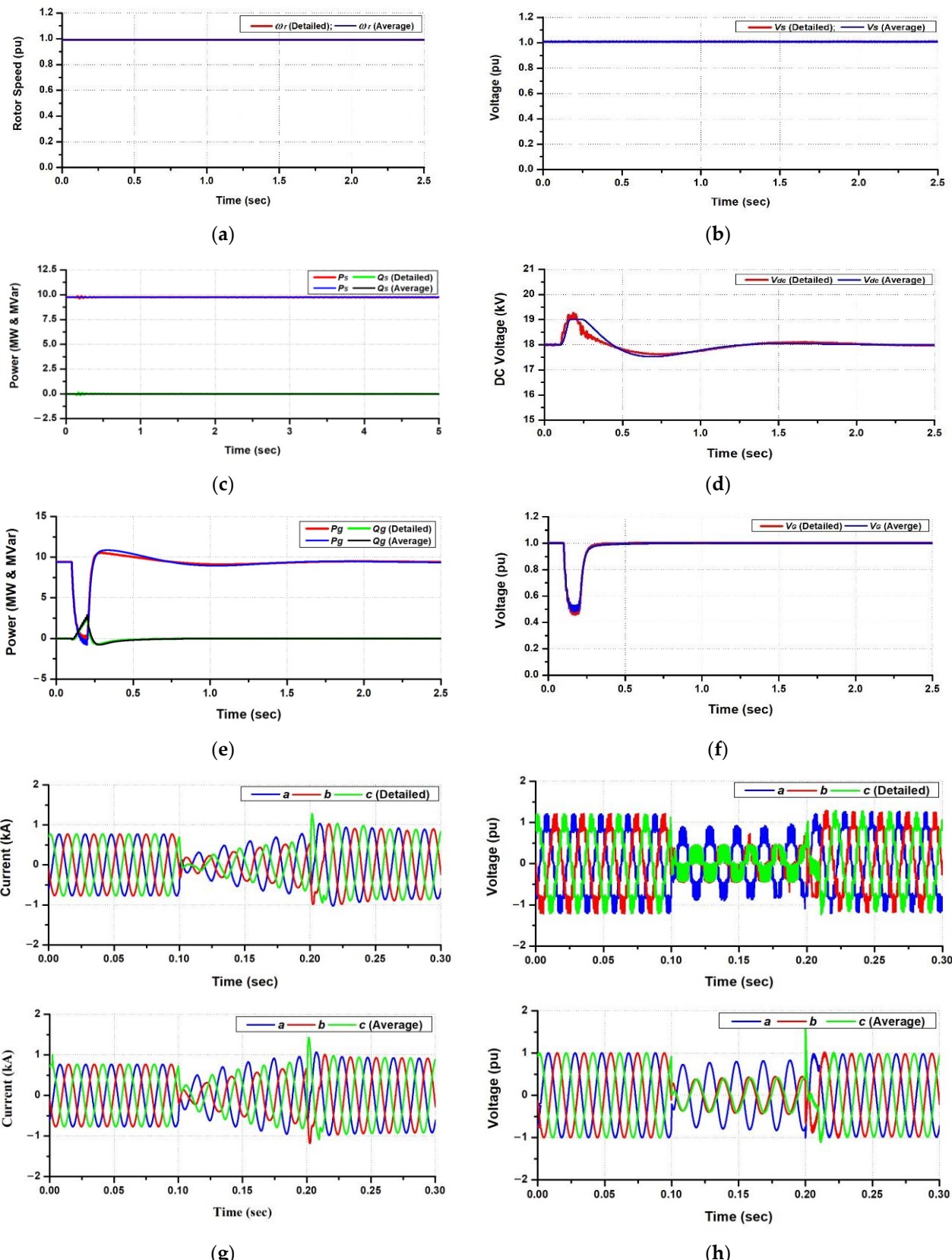

**Figure 22.** Dynamic Performances of WTG1 in the case of 2LG: (**a**) Rotor speed response; (**b**) Voltage profiles at the terminal of the stator winding; (**c**) Active and reactive power output of the PMSG; (**d**) DC link circuit voltage profile; (**e**) Active and reactive power output of grid-side MMC; (**f**) Voltage profile at Bus BG1; (**g**) Waveform of current at the grid-side MMC; (**h**) Waveform of voltage at the grid-side MMC.

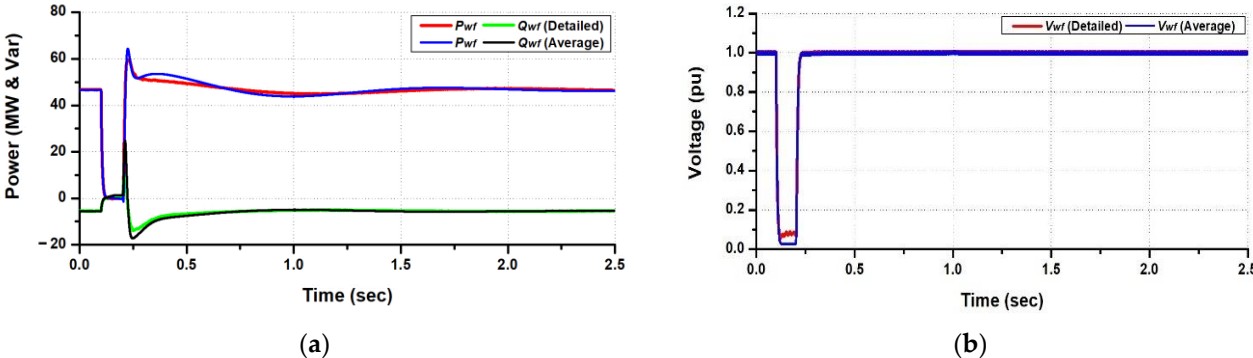

**Figure 23.** Power output and voltage performances of the wind farm at Bus B33 under the 3LG case: (**a**) Active power output of wind farm; (**b**) Voltage profile at Bus B33.

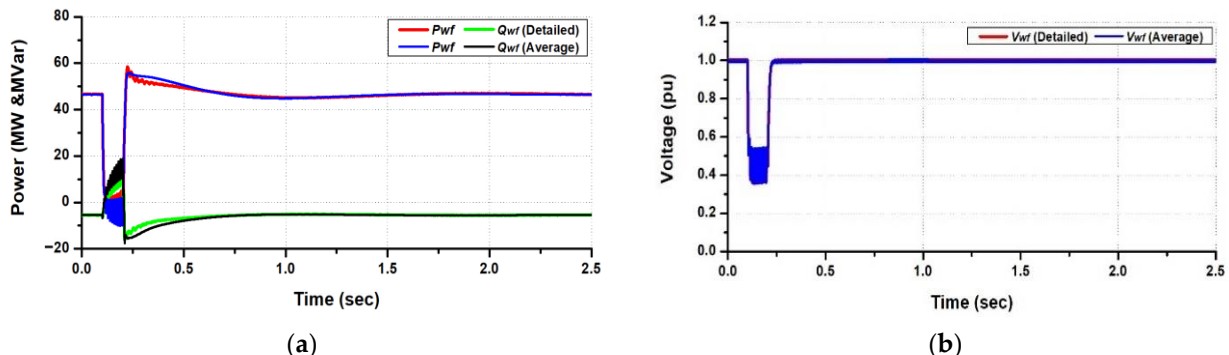

**Figure 24.** Power output and voltage performances of the wind farm at Bus B33 under the 2LG case: (**a**) Active power output of wind farm; (**b**) Voltage profile at Bus B33.

## 5. Conclusions

The dynamic performances of the proposed detailed and the proposed average models of a grid-connected MMC-controlled permanent magnet wind turbine generator have been investigated. Comparative analyses between the detailed and the average models have been performed under steady-state and transient conditions, and it has been confirmed that both models have almost the same dynamic responses and have a good controllability. Although both models can be used in the analysis of grid-connected wind farms, each model can have different purposes in applications.

The proposed detailed model requires a small discrete time step because power electronics IGBTs and their switching phenomena are considered. The detailed model is suited to analyzing the dynamic performance in the control system design of an individual wind turbine generator in a short time simulation. The dynamic performance of an MMC's sub-modules and harmonics analyses can be handled by the detailed model.

In the proposed average model, power electronics IGBTs converter and switching phenomena are omitted, and thus the simulation time step can be much larger than that of the detailed model. The model can be used as an individual or an aggregated model of wind turbine generators and wind farms. The model is suited to analyzing a power system with many wind generators, which cannot be analyzed using the detailed model.

In the future, the development of control strategies to improve the dynamic performance of the MMC in controlling the power flow from the generator to the grid system will be performed. Improvement of the gain control by adopting linear or nonlinear methods would be one of the key points in the next research. Optimization methods based on Artificial Intelligence (AI) will also be considered in future research.

**Author Contributions:** Conceptualization, M.R. and J.T.; software, M.R.; supervision, A.U., R.T. and J.T.; validation, A.U., R.T. and J.T.; writing—original draft, M.R. and J.T. All authors have read and agreed to the published version of the manuscript.

**Funding:** This research has not received any external funding.

**Institutional Review Board Statement:** Not applicable.

**Informed Consent Statement:** Not applicable.

**Data Availability Statement:** Not applicable.

**Conflicts of Interest:** The authors declare no conflict of interest.

## Appendix A

In this paper, it should be noted that a 3-phase *abc* to *dq*0 transformation as well as its inverse are used, and they are expressed by the following equations [22]:

Park Transformation

$$
\begin{bmatrix} d \\ q \\ 0 \end{bmatrix} = \frac{2}{3} \begin{bmatrix} \cos(\theta) & \cos\left(\theta - \frac{2\pi}{3}\right) & \cos\left(\theta + \frac{2\pi}{3}\right) \\ \sin(\theta) & \sin\left(\theta - \frac{2\pi}{3}\right) & \sin\left(\theta + \frac{2\pi}{3}\right) \\ \frac{1}{2} & \frac{1}{2} & \frac{1}{2} \end{bmatrix} \begin{bmatrix} a \\ b \\ c \end{bmatrix} \tag{A1}
$$

Invers Park Transformation

$$
\begin{bmatrix} a \\ b \\ c \end{bmatrix} = \frac{2}{3} \begin{bmatrix} \cos(\theta) & \sin(\theta) & 1 \\ \cos\left(\theta - \frac{2\pi}{3}\right) & \sin\left(\theta - \frac{2\pi}{3}\right) & 1 \\ \cos\left(\theta + \frac{2\pi}{3}\right) & \sin\left(\theta + \frac{2\pi}{3}\right) & 1 \end{bmatrix} \begin{bmatrix} d \\ q \\ 0 \end{bmatrix} \tag{A2}
$$

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
