# Peer review of "Detailed and Average Models of a Grid-Connected MMC-Controlled Permanent Magnet Wind Turbine Generator"

_applsci, doi:10.3390/app12031619_

Round 1
Reviewer 1 Report
In this paper detailed model and average model of the Modular Multilevel Converter (MMC) controlled wind turbine direct driven Permanent Magnet Synchronous Generator (PMSG) is presented. The aim of modeling is to analyze dynamic performances of the grid connected wind turbine generator. Thus the topic of the paper is interesting and actual. It suites to Applied Sciences journal scope.
The literature review in this paper is based on 26 positions. They are adequate to article content. But there is a luck of actual positions. Thus my recommendation #1 is to extend the literature review with 5 positions from 2020-2021 year.
The introduction section is correct - general information is given with the main contribution of the paper. But my recommendation #2 is to add at the end of the introduction a separate paragraph with short summary of paper organization. Please use "Secion 2 presents ..., section 3 concerns ... etc"
For section 2 that presents " Detailed Model of MMC Controlled Wind Turbine Generator" I have no recommendation to improve this part. It's clear and correct.
For section 3 that presents " Average Model of MMC Controlled Wind Turbine Generator" I have no recommendation to improve this part. It's clear and correct.
For section 4 "Simulation and Analysis" I have recommendation #3. Please add as subsection "4.3 Discussion" where the discussion of results and simulations will be performed. It's worth validating those results with similar works in literature or just confirming their correctness.
For section 5 "Conclusions: I have recommendation #4 that concerns future research directions. Please add a paragraph with future research directions.
Recommendation #5 concerns technical issues. Please add essential parts of the article such as author contribution, author statement, conflict of interest, etc. Please use the Applied Science temple.
Reviewer 2 Report
This paper describes the comparison between the detailed model and the average model of the MMC converter for a wind turbine generator.
However, the proposed method only replaces the MMC converter with a voltage source, and its novelty and usefulness are not recognized.
-Chap.4: It is easy to assume that there is almost no difference between the detailed model and the average model in the phenomenon targeted in the simulation case. In the phenomenon of time in the simulation case, the problem of the average model should be clarified.
-Section 4.2: The detailed model is often used when performing waveform level analysis. A comparison of voltage and current waveforms should also be shown in the transient analysis in Section 4.2. Furthermore, in Fig. 21 and Fig. 22, I think it is necessary to consider in detail the difference between the results of the two models when the CBs are opened.
- In the case of an unbalanced fault such as a two-phase short circuit, the capacitor voltage of the sub-module of each arm will be different. I think this paper should be added a case of unbalanced fault to the simulation case.
Reviewer 3 Report
In this paper detailed model and average model of the MMC-controlled wind turbine direct driven PMSG are proposed. To further improve this paper, my comments are given as follows. 1. Either MMC or PMSG has been investigated for many years. What are the advantages of the proposed two models compared with the existing models? Authors should clarify this issue. 2. The literature review for wind turbine could be enriched [1,2]. 1) Energy-shaping L2-gain controller for PMSG wind turbine to mitigate subsynchronous interaction, Electrical Power and Energy Systems. 2) Sliding mode controller based on feedback linearization for DFIG-based wind power plants. 3. In fact, a practical wind farm is usually composed of hundreds of wind turbine generators (WTGs). Could the proposed damping controller achieve satisfactory performance in a practical wind farm? 4. It seems that both proposed models neglect the inherent nonlinearity of MMC-controlled PMSG. Authors should demonstrate the effectiveness of the propose models under different operation conditions.Author Response
Please see the attachment

Round 2
Reviewer 1 Report
Dear Authors,
The manuscript is suitable for publication after the revision.
Best regards,
Reviewer
Reviewer 2 Report
This paper describes the comparison between the detailed model and the average model of the MMC converter for a wind turbine generator. I think the proposed method is useful for large-scale wind farm analysis.
As future research, since there is a difference in the distortion of the voltage waveforms between the two models, I expect a detailed analysis of the voltage waveforms.